# Availability and use of personal protective equipment in low- and middle-income countries during the COVID-19 pandemic

Salomé Henriette Paulette Drouard [1]*, Tashrik Ahmed[2], Pablo Amor Fernandez[1], Prativa Baral [2], Michael Peters [2], Peter Hansen[2], Tawab Hashemi[2], Isidore Sieleunou[2], Munirat Iyabode Ogunlayi[2], Alain-Desire Karibwami[2], Julie Ruel Bergeron[2], Edwin Eduardo Montufar Velarde[3], Mohamed Lamine Yansane[4], Chea Sanford Wesseh[5], Charles Mwansambo[6], Charles Nzelu[7], Helal Uddin[8], Mahamadi Tassembedo [9], Gil Shapira[1]

1 World Bank Group, Washington, DC, United States of America, 2 Global Financing Facility for Women, Children, and Adolescents, 3 Ministerio de Salud Pública y Asistencia Social, Guatemala, 4 Ministry of Health and Public Hygiene, Guinea, 5 Ministry of Health and Social Welfare, Liberia, 6 Ministry of Health, Lilongwe, Malawi, 7 Federal Ministry of Health, Abuja, Nigeria, 8 Directorate General of Health Services, Dhaka, Bangladesh, 9 Ministère De La Santé, Kadiogo, Burkina Faso

* drouard.salome@gmail.com

**Data Availability Statement:** The data underlying this article were provided by and are property of the ministries of health of the seven countries

## Abstract

### Background

Availability and appropriate use of personal protective equipment (PPE) is of particular importance in Low and Middle-Income countries (LMICs) where disease outbreaks other than COVID-19 are frequent and health workers are scarce. This study assesses the availability of necessary PPE items during the COVID-19 pandemic at health facilities in seven LMICs.

### Methods

Data were collected using a rapid-cycle survey among 1554 health facilities in seven LMICs via phone-based surveys between August 2020 and December 2021. We gathered data on the availability of World Health Organization (WHO)-recommended PPE items and the use of items when examining patients suspected to be infected with COVID-19. We further investigated the implementation of service adaptation measures in a severe shortage of PPE.

### Results

There were major deficiencies in PPE availability at health facilities. Almost 3 out of 10 health facilities reported a stock-out of medical masks on the survey day. Forty-six percent of facilities did not have respirator masks, and 16% did not have any gloves. We show that only 43% of health facilities had sufficient PPE to comply with WHO guidelines. Even when all items were available, healthcare workers treating COVID-19 suspected patients were reported to wear all the recommended equipment in only 61% of health facilities. We did not

participating in the analysis. The data will be shared on reasonable request with permission of the seven ministries from gffsecretariat@worldbank. org.

**Funding:** The author(s) received no specific funding for this work.

**Competing interests:** The authors declare no competing interest.

find a statistically significant difference in implementing service adaptation measures between facilities experiencing a severe shortage or not.

## Conclusion

After more than a year into the COVID-19 pandemic, the overall availability of PPE remained low in our sample of low and middle-income countries. Although essential, the availability of PPE did not guarantee the proper use of the equipment. The lack of PPE availability and improper use of available PPE enable preventable COVID-19 transmission in health facilities, leading to greater morbidity and mortality and risking the continuity of service delivery by healthcare workers.

## Introduction

Ensuring the availability and use of personal protective equipment (PPE) among healthcare workers (HCWs) is essential for reducing the transmission of infectious diseases within health facilities. The 2013–2016 West Africa Ebola epidemic demonstrated that HCW mortality and morbidity weakens the capacity for crisis response and created long-term challenges in providing primary health services [1]. The use of PPE is an important strategy to protect HCW and patients from the spread of pathogens and cross-contamination [2, 3]. During the COVID-19 pandemic, the prevalence of infection is disproportionately higher among HCW than in the general population: HCW represented less than 3% of the global population but in 2020 accounted for more than 14% of the infections [3]. Though PPE supply chains had stabilized, insufficient PPE availability was a major source of service disruption in 26% of countries between January 2021 and March 2021 [4]. Improving PPE availability and use is a cost-effective and straightforward way to protect the health workforce during epidemics [5].

In March 2020, World Health Organization guidance defined appropriate PPE for interacting with a suspected COVID-19 patient as the use of: "*a medical mask [. . .] [and] eye protection (goggles) or facial protection (face shield) to avoid contamination of mucous membranes; [. . .] [and] a clean, non-sterile, long-sleeved gown; [. . .] [and] gloves*" [6]. Additional airborne precautions must be taken by wearing a respirator (e.g., N95 or FFP2) for aerosol-generating procedures such as intubation or noninvasive ventilation.

In low and middle-income countries (LMICs), shortages and non-compliance to guidelines on PPE use pre-dated the COVID-19 pandemic [7]. For example, severe shortages of face masks were documented in the Service Provisions Assessments (SPAs) in the Democratic Republic of Congo (2018), Nepal (2015), and Tanzania (2015) [8]. Moreover, a systematic review on PPE use for respiratory infections from 2019 emphasized the low level of compliance with PPE use among HCWs in Pakistan [9]. Despite efforts to strengthen PPE supply during the pandemic, initial evidence suggested that global shortages had persisted. For instance, data from a facility phone survey in Kenya in July 2021 showed that only 15% had access to the complete PPE set available at the health facility [10].

In response to global stock-outs of PPE and to limit the spread of the virus among HCW, WHO recommended adapting service provision when severe shortages were experienced by limiting face-to-face interaction between HCWs and patients [11]. Extending prescriptions, encouraging self-care, providing all care in a single visit, and switching to a digital platform are relevant to service adaptions to respond to a severe PPE shortage as recommended by WHO.

The lack of equipment, combined with low capacities to adapt service delivery, increases the risk of HCWs infection and limits the response to epidemics [12]. To our knowledge, the implementation of infection prevention and control (IPC) measures and service adaptation in LMICs in response to COVID-19 have not yet been measured. Documenting the implementation of IPC measures and service adaptation is essential to highlight strategies to ensure the safety of HCWs and the continuity of essential health services during prolonged and future PPE shortages.

There is limited recent evidence in LMICs on PPE availability and use since the early pandemic or on the implementation of service adaptation in response to experiencing a severe shortage. In this paper, we described the availability of COVID-19 appropriate PPE in seven LMICs during the pandemic and the use of these barriers by HCWs when providing care to suspected and confirmed cases of COVID-19. In cases of severe PPE shortage, we further assessed the implementation of service adaptation measures.

## Methods

### Overview and sample selection

To monitor the continuity of essential health services during the pandemic, the Global Financing Facility for Women, Children, and Adolescents (GFF) supported partner countries in implementing rapid-cycle phone-based health facility surveys. In this context, implementation of facility phone surveys was offered to all partner countries. The seven countries covered by this study are the ones that opted to implement the phone survey and for which at least one round of data was completed by August 2021. These surveys assessed the effect of the pandemic on the ability to deliver essential health services and document adaptations to service delivery modalities. Surveys were conducted in Bangladesh, Burkina Faso, Guatemala, Guinea, Liberia, Malawi, and Nigeria between May 2021 and August 2021. All samples, besides Nigeria, are nationally representative and stratified by administrative units. From a master facility list provided by the Ministry of Health, Health facilities were randomly selected within each administrative unit, and the number of health facilities picked reflects the weight of the stratum at the national level. The Nigeria sample was stratified by the COVID burden at the state level as of August 2020, S1 Table details the number of rounds and reference periods for each country. Standard questionnaires were adapted to each country's context and priorities. The specific sampling strategy varied by country and is presented in S2 Table.

### Data collection

Survey respondents generally included facility officer in-charges, but in some cases other respondents, like facility administrators, were better suited to answer modules within the survey. Three attempts were made to reach each facility, and interview times were scheduled in advance to minimize burden on the respondents. In case of non-response, a replacement facility of the same facility level in the same province was randomly selected from the list of eligible health facilities when possible. More details on the response rate are available in S3 Table. All the health facility representatives we managed to reach accepted to take part in the survey.

### Analysis

To assess availability, we computed the frequencies of health facilities reporting the availability of any PPE within health facilities, the availability of PPE to all healthcare workers, and the availability of a complete PPE set as defined by WHO [6]. We examined the availability of the following PPE: gowns, goggles, face shields, gloves, medical masks, and respirators (N95 of

FFP2). Availability is described by 1) the presence of at least one of each type of PPE within the health facility and 2) the availability of each PPE to all health workers. As defined by March 2020 WHO guidance [6], we measured the frequency of the availability of a complete PPE set as composed of a gown, a pair of gloves, face or eye protection, and a mask (medical or respirator).

Our study investigated the use of PPE when examining COVID-19 suspected patients. Use of PPE was assessed by a self-report of the PPE health workers routinely used during a consultation with a suspected or confirmed COVID-19 case. This is benchmarked against the set of PPE recommended by WHO guidance; i.e., HCWs wearing a protective gown, eye or facial protection (goggles or face shields), gloves, and a mask (medical masks or respirators).

Finally, we considered health facility service adaptation in the event of a severe shortage of PPE barriers [11]. There is not a unique definition of severe PPE shortage. We chose to define health facilities without any available gloves or masks (medical masks and respirators) as experiencing a severe shortage. Gloves and masks are the minimum set of required PPE to maintain spatial separation for basic contact and droplet precautions for healthcare workers caring for suspected COVID-19 patients. Gloves and masks are also more difficult to replace with an alternative or homemade PPE. In facilities with severe shortages, adaptations to service delivery to limit in-person consultations according to WHO guidance were assessed by four possible service adaptation measures: extending prescriptions, encouraging self-care, providing all care in a single visit, and switching to a digital platform. We also investigated the adoption of different Infection Prevention Control (IPC) measures health facilities took during a severe shortage of PPE. We considered different IPC measures to respond to COVID-19, such as regular cleaning of surfaces, available hand-washing stations and a dedicated entry for staff members, screening patients for COVID-19, implementing a triage system with COVID-19 dedicated areas, and maintaining social distancing. To understand if service adaption reflects PPE severe shortage, we analyzed the likelihood of health facilities adopting each service adaptation and IPC measure when experiencing a stock out. The likelihood of adopting each measure when experiencing a severe shortage was assessed by $X^2$ tests.

### Ethical approval

The study was requested, reviewed, and approved by a director-level official in each Ministry of Health and was exempted from human subjects research as public health practice in every country except Burkina Faso. In Burkina Faso, ethical approval was received from the ethics committee of the local author's institute. Survey participation was voluntary and verbal consent was received from all respondents.

## Results

### Sample characteristics

The total sample included 1554 health facilities from seven countries (Table 1). Seventy-two percent of the health facilities were rural, 7% were peri-urban, and 21% urban. Health facilities were either hospitals, health centers, or health posts defined by the country's health management system. Forty-nine percent of the health facilities were health centers. Eighty-six percent were from the public sector.

### Availability assessment

There were major deficiencies in PPE availability at health facilities (Table 2), as well as substantial variation across items, countries, and facility types. Shortages existed for all PPE items.

**Table 1. Facility characteristics.**

| | Bangladesh (n = 291) | Burkina Faso (n = 159) | Guatemala (n = 239) | Guinea (n = 156) | Liberia (n = 116) | Malawi (n = 192) | Nigeria (n = 401) | Total (n = 1554) |
|---|---|---|---|---|---|---|---|---|
| **Location** | | | | | | | | |
| *Urban (%)* | 20 | 10 | 24 | 29 | 21 | 7 | 27 | 21 |
| *Peri-urban (%)* | 17 | 2 | 0 | 0 | 0 | 5 | 10 | 7 |
| *Rural (%)* | 63 | 88 | 75 | 71 | 79 | 88 | 63 | 72 |
| **Facility type** | | | | | | | | |
| *Hospital (%)* | 33 | 3 | 4 | 3 | 12 | 8 | 10 | 12 |
| *Health center (%)* | 33 | 97 | 15 | 94 | 10 | 84 | 31 | 47 |
| *Health Post/ Clinic (%)* | 34 | 0 | 81 | 3 | 78 | 8 | 55 | 40 |
| *Other (%)* | 0 | 0 | 0 | 0 | 0 | 0 | 3 | 1 |
| **Managing authority** | | | | | | | | |
| *Government, public (%)* | 100 | 100 | 100 | 100 | 100 | 72 | 60 | 86 |
| *Private, for profit (%)* | 0 | 0 | 0 | 0 | 0 | 1 | 40 | 11 |
| *Private, nonprofit (%)* | 0 | 0 | 0 | 0 | 0 | 27 | 0 | 3 |

Table 2 shows that only 43% of health facilities had sufficient PPE available to comply with WHO guidelines on the day of the survey. Almost 3 out of 10 health facilities reported a stock-out of medical masks on the day of the survey. Forty-six percent of facilities did not have respirator masks, and 16% did not have any gloves. On average, health facilities in our sample had 4.1 types of PPE available out of the six recommended during the COVID-19 pandemic. Facilities in Bangladesh and Guinea had the lowest availability of all items, with an average of 2.7 items out of six. At the other end of the spectrum, Liberian facilities, on average, reported 5.3 items available. In all countries, hospitals had a higher average availability of items in comparison to the primary-level facilities. The availability of PPE was near 100% for hospitals in Malawi, Liberia, Guatemala, and Burkina Faso. In countries where several rounds of data collection took place, we did not observe substantial changes in the availability of supplies between February 2021 and August 2021 (S1 Table), with only a one percentage point average change between the first and last round.

Even in countries where PPE was generally available at the facility level, there were often insufficient quantities to protect all health workers, as shown in Fig 1. For example, although 78% of health facilities reported having masks in Liberia, only 38% of facilities had enough masks for all HCW. For medical masks, Guinea had the lowest availability for all HCW at 26%. For respiratory masks, the average availability across all countries was 43% and was lowest in Bangladesh at 7%.

## Compliance with WHO guidelines on PPE with COVID-19 suspected cases

Regarding the different PPE barriers used when examining suspected COVID-19 cases, HCWs were reported to wear masks (medical or respirators) in 80% and gloves in 85% of health facilities of the full sample, as shown in Table 3. Eye or facial protection was the least likely recommended PPE to be worn (65%).

We also found that HCWs did not use all the recommended PPE barriers when examining COVID-19 suspected or confirmed cases even when all items are available at the health facility.

**Table 2. Availability at the health facility level for each piece of PPE by country by facility type.**

| Country | Facility type | n | # items | sd | Gown | Gloves | Goggles | Face shields | N95/FFP2 | Medical masks | Complete PPE set |
|---------|---------------|---|---------|-----|------|--------|---------|-------------|----------|---------------|------------------|
| Bangladesh | Total | 291 | 2.7 | 2.25 | 51% | 55% | 42% | 38% | 25% | 63% | 28% |
| | Hospitals | 96 | 4.9 | 1.36 | 90% | 95% | 82% | 73% | 53% | 94% | 67% |
| | Health centers | 96 | 1.9 | 2.06 | 33% | 43% | 28% | 26% | 14% | 47% | 16% |
| | Health posts | 99 | 1.5 | 1.56 | 31% | 28% | 16% | 17% | 8% | 49% | 3% |
| Burkina Faso | Total | 159 | 4.7 | 1.07 | 94% | 97% | 79% | 91% | 59% | 55% | 62% |
| | Hospitals | 4 | 5.8 | 0.50 | 100% | 100% | 100% | 100% | 100% | 75% | 100% |
| | Health centers | 155 | 4.7 | 1.07 | 94% | 97% | 79% | 90% | 58% | 54% | 61% |
| Guatemala | Total | 239 | 4.9 | 1.34 | 79% | 93% | 82% | 79% | 72% | 89% | 60% |
| | Hospitals | 10 | 5.9 | 0.32 | 100% | 100% | 100% | 100% | 90% | 100% | 100% |
| | Health centers | 35 | 5.5 | 1.04 | 89% | 97% | 91% | 91% | 86% | 97% | 77% |
| | Health posts | 194 | 4.8 | 1.38 | 77% | 92% | 80% | 76% | 68% | 87% | 55% |
| Guinea | Total | 156 | 2.7 | 1.67 | 64% | 63% | 47% | 52% | 10% | 31% | 8% |
| | Hospitals | 5 | 1.8 | 2.05 | 40% | 20% | 40% | 40% | 0% | 40% | 20% |
| | Health centers | 146 | 2.7 | 1.66 | 63% | 65% | 45% | 52% | 10% | 30% | 8% |
| | Health posts | 5 | 1.2 | 1.79 | 40% | 20% | 40% | 20% | 0% | 0% | 0% |
| Liberia | Total | 116 | 5.3 | 1.05 | 97% | 97% | 85% | 91% | 80% | 78% | 72% |
| | Hospitals | 14 | 5.6 | 0.84 | 100% | 86% | 93% | 100% | 93% | 93% | 86% |
| | Health centers | 11 | 5.4 | 0.81 | 100% | 100% | 82% | 100% | 82% | 73% | 64% |
| | Health posts | 91 | 5.2 | 1.09 | 96% | 98% | 85% | 89% | 78% | 77% | 71% |
| Malawi | Total | 192 | 4.8 | 1.48 | 68% | 95% | 73% | 88% | 74% | 97% | 53% |
| | Hospitals | 15 | 5.9 | 0.35 | 93% | 93% | 100% | 100% | 100% | 100% | 87% |
| | Health centers | 161 | 4.7 | 1.53 | 63% | 94% | 68% | 87% | 70% | 96% | 47% |
| | Health posts | 16 | 5.8 | 0.58 | 94% | 100% | 100% | 94% | 88% | 100% | 88% |
| Nigeria | Total | 401 | 4.0 | 1.76 | 70% | 95% | 45% | 50% | 61% | 80% | 36% |
| | Hospitals | 42 | 4.7 | 1.27 | 93% | 98% | 50% | 64% | 76% | 90% | 48% |
| | Health centers | 126 | 4.0 | 1.62 | 75% | 95% | 45% | 48% | 60% | 73% | 33% |
| | Health posts | 220 | 3.9 | 1.89 | 64% | 94% | 44% | 49% | 56% | 85% | 36% |
| | Other | 13 | 3.8 | 1.59 | 62% | 100% | 38% | 38% | 100% | 38% | 23% |
| **Sample** | | **1554** | **4.1** | **1.9** | **72%** | **84%** | **60%** | **65%** | **54%** | **73%** | **43%** |

Restricting the sample to only health facilities with the complete PPE set available, we found that HCWs were wearing all the recommended barriers in only 61% of health facilities. The percentage was as low as 47% in Nigeria. HCWs were reported to wear protective gowns and gloves in 82% and 91% of health facilities when examining suspected or confirmed COVID-19 cases. Almost two health facilities out of 10 reported their HCWs skipped using masks or respirators in such cases, although the equipment was available. When asked about appropriate PPE use, health facility representatives reported their staff always used PPE correctly in less than half of the facilities with all pieces available, with minimal variation across countries.

## Service adaptation and IPC measures when experiencing a severe shortage

We then explored whether facilities with PPE shortages implement the service adaptation measures recommended by the WHO guideline. Primarily, according to our definition of a severe shortage (i.e., neither mask nor gloves were available within the facility), 23% (229) of the health facilities were experiencing a severe shortage on the survey day as shown in Table 2.

IPC measures were generally more implemented than service delivery adaptations. We observed that 86% of the health facilities provided additional hand-washing stations, and 83%

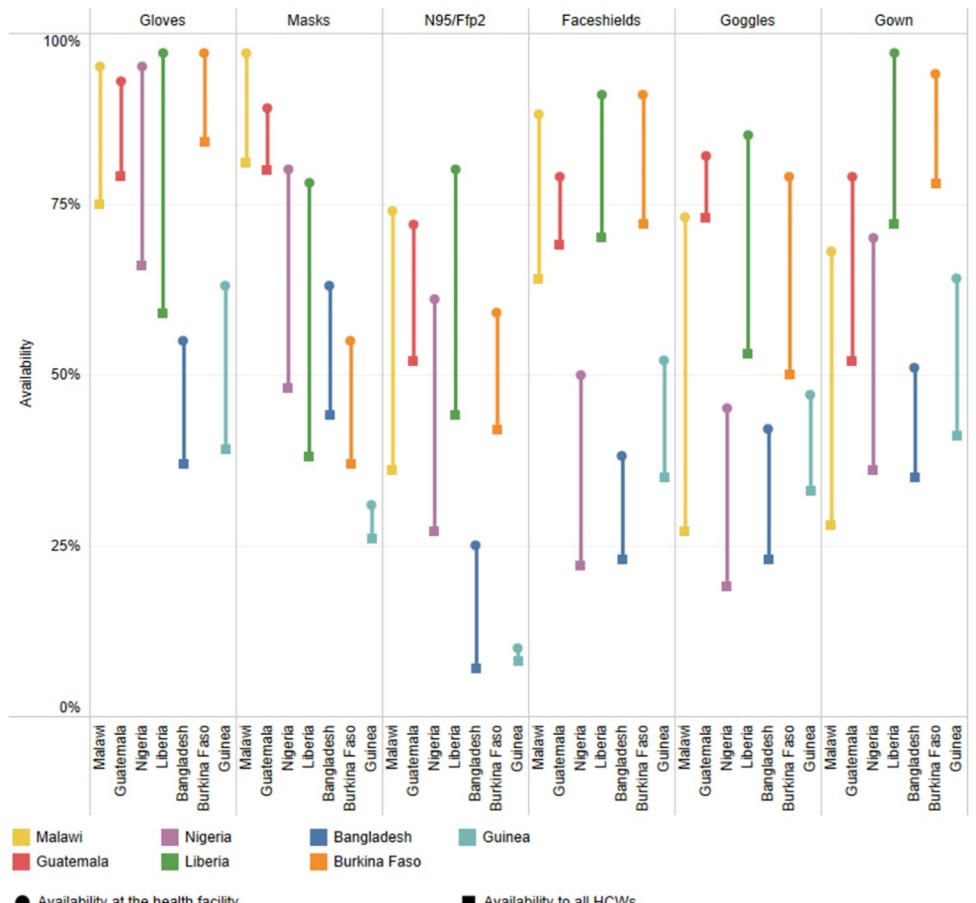

**Fig 1. Availability at the HF level and to all HCWs for each piece of PPE by country.** The gap in availability is shown by the distance between the availability of the PPE barrier at the health facility and the availability to all HCWs points.

implemented social distancing, while 60% encouraged self-care and 58% provided all care in a single visit (Table 4).

Implementing service adaptations can reduce the health risk during an in-person visit to the health facility when PPE is lacking. However, we did not observe a statistical difference in

**Table 3. PPE use: Barriers used when examining COVID-19 suspected patients and correct use of the equipment.**

| Country | n | Gowns | Gloves | Eye/facial protection | Mask or respirator | Wear the complete PPE set | Always use PPE correctly |
|---|---|---|---|---|---|---|---|
| **Bangladesh** | 291 | 52% | 74% | 55% | 98% | 35% | 44% |
| **Burkina Faso** | 159 | 91% | 92% | 94% | 72% | 67% | 40% |
| **Guinea** | 156 | 78% | 92% | 79% | 71% | 49% | 37% |
| **Nigeria** | 401 | 66% | 87% | 56% | 74% | 30% | 48% |
| *Sample* | *1007* | 68% | 85% | 65% | 80% | 41% | 44% |
| *Only for health facilities with a complete PPE set available* | | | | | | | |
| **Bangladesh** | 103/291 | 70% | 94% | 86% | 100% | 66% | 42% |
| **Burkina Faso** | 120/159 | 93% | 95% | 96% | 80% | 76% | 42% |
| **Guinea** | 18/156 | 94% | 94% | 94% | 94% | 83% | 39% |
| **Nigeria** | 178/401 | 80% | 87% | 75% | 73% | 47% | 58% |
| *Sample* | *419/1007* | 82% | 91% | 84% | 83% | 61% | 49% |

**Table 4. Service adaptation and IPC measures.**

| | Experiencing shortage | | Not experiencing shortage | | Full sample | | p-values | |
|---|---|---|---|---|---|---|---|---|
| | n | % | n | % | n | % | | Countries in the sample |
| *IPC measures* | | | | | | | | |
| Regular surface cleaning | 239 | 68% | 556 | 85% | 795 | 79% | 0.0000 | Bangladesh, Burkina Faso, Guatemala, Guinea, Liberia, Malawi, Nigeria |
| Hand washing stations | 294 | 77% | 779 | 90% | 1073 | 86% | 0.0000 | Bangladesh, Burkina Faso, Guatemala, Guinea, Nigeria |
| Specific staff entrance | 133 | 32% | 635 | 56% | 768 | 50% | 0.0000 | Bangladesh, Burkina Faso, Guatemala, Guinea, Liberia, Malawi, Nigeria |
| Screening patients for COVID-19 | 167 | 42% | 718 | 69% | 885 | 62% | 0.0000 | Bangladesh, Burkina Faso, Guatemala, Guinea, Malawi, Nigeria |
| Triage system for patients | 131 | 33% | 656 | 63% | 787 | 55% | 0.0000 | Bangladesh, Burkina Faso, Guatemala, Guinea, Malawi, Nigeria |
| Social- distancing | 139 | 77% | 319 | 86% | 458 | 83% | 0.0070 | Bangladesh, Liberia, Malawi, Nigeria |
| *Service adaptation* | | | | | | | | |
| Extend prescription | 70 | 31% | 331 | 46% | 401 | 42% | 0.0001 | Bangladesh, Liberia, Malawi, Nigeria |
| Encourage self-care | 148 | 65% | 421 | 58% | 569 | 60% | 0.0574 | Bangladesh, Liberia, Malawi, Nigeria |
| Provide all care in a single visit | 136 | 60% | 415 | 58% | 551 | 58% | 0.4863 | Bangladesh, Liberia, Malawi, Nigeria |
| Switch to digital platform | 49 | 25% | 124 | 17% | 173 | 27% | 0.4087 | Bangladesh, Nigeria |

the implementation of such adaptations whether or not health facilities were experiencing a severe shortage. Thirty-one percent of the health facilities experiencing a severe shortage chose to extend prescription periods, 65% encourage self-care, 60% combine different services in a single visit, and 25% switch to digital platforms (Table 4). We observed similar magnitudes for implementing these measures in health centers not experiencing a severe shortage: 46% extended prescription, 58% encouraged self-care, 58% provided all care in a single visit, and 17% switched to digital platforms.

We found a significant positive correlation between PPE availability and adopting standard and COVID-19 specific IPC measures. Almost all (90%) health facilities with masks and gloves available had hand-washing stations inwards, compared to only 77% of health facilities experiencing a severe shortage. Health facilities not experiencing a severe PPE shortage were more likely to ensure social distancing was maintained within the facility by nine percentage points.

We also tested for different definitions of severe shortage, such as no mask available within the facility and having less than 3 of the necessary PPE pieces for a complete set. Changing the definition of PPE shortage did not affect the lack of correlation between shortages and service delivery adaptation. We ran this analysis differentiating by facility type and country, no significant relationship was found.

## Discussion

We found that, after more than a year into the COVID-19 pandemic, most health facilities in LMICs were not fully equipped to respond to the COVID-19 pandemic. PPE availability was notably low in Guinea, Bangladesh, and Nigeria, where fewer than 70% of health facilities have all the recommended PPE. The shortage was particularly severe for respirators and masks. Less than half of all the health facilities sampled had medical masks available for all HCWs on the survey day. Hospitals had greater availability of the different PPE items in almost all settings. N95 or FFP2 respirators were only available to 15% of the health facilities in Guinea and 25% of the facilities in Bangladesh. These results were consistent with other studies on PPE availability in LMICs. For instance, the United Nations Office for Disaster Risk Reduction facility assessment in Kenya highlighted similar results. The complete PPE set was only available in

64% of the health facilities, and when items were available, stocks were usually too low to supply all HCWs [10]. We also observed substantial differences in PPE use across countries. This discrepancy can reflect many country-specific factors such as the existence of a domestic supplier, the strength of supply chains, health worker awareness/training, and/or the lack of global quality standards on the equipment [13]. Despite international donor, multilateral agency, government, and industry efforts to rapidly procure affordable and safe PPE during the early stages of the COVID-19 pandemic, many health facilities in LMICs had limited availability of PPE [14, 15]. While exacerbated by the pandemic, these shortages could have been impacted by pre-existing conditions.

Beyond the current pandemic context, PPE shortage is a chronic issue hindering the capacity to provide care in LMICs. Although we reported PPE shortages across contexts, availability might be better than in the pre-pandemic period in some settings. Gage, A., and Bauhoff, S. (2020) used the Service Provision Assessments (SPAs) of seven LMICS from 2015 to 2018 and found that face masks were available in less than a third of non-hospitals in Bangladesh, DRC, Nepal, and Tanzania [8]. The average availability of face masks in lower-level structures in our sample was close to 56%, and only in Guinea is availability lower than a third. SPA data from 2017 in Bangladesh also show that medical masks were only available in 28% of health facilities compared to 63% in our study [8]. The self-reported data on PPE availability before the pandemic in Burkina Faso and Guinea presented in S4 Table corroborates this idea. The increased availability compared to pre-pandemic data may reflect the effort to procure the equipment during the first months of the pandemic. However, the stagnation of the level of availability over rounds indicates persisting gaps in availability at many health facilities. Integrating global standards and availability targets for PPE into the health system's preparedness evaluation may also incentivize countries to increase supply [13]. But, increasing supply alone may not be sufficient to protect health workers.

We also found that HCWs examining COVID-19 suspected cases were not systematically using the complete PPE set even when all barriers were available in their facilities. Many factors might explain why HCWs were not using PPE. PPE is constraining. Studies show that wearing PPE increases heat stress during practice and reduces HCWs' performance [16]. HCWs may or may not wear PPE based on their risk perception. The occupational hazards may be perceived as less acute when patients do not exhibit physical symptoms [17]. In Malaysia, infection among HCWs was primarily driven by the inappropriate use of PPE when examining asymptomatic patients [18]. Incomplete knowledge, low level of training, and negative perception of equipment can also increase non-compliance with PPE protocols [19]. Additionally, facility-level interventions and policies to preserve PPE and encourage use may have an impact on the use of PPE.

Although there were country-specific guidelines and different levels of risk during a consultation, we did not find significant evidence of health facilities actively implementing measures to minimize the risk of infection in the event of a severe PPE shortage. Moreover, our results suggested health facilities with greater availability of PPE were more likely to implement IPC measures. Health facilities may have different priorities and resources available for IPC and service delivery adaptation [20]. The high risk of nosocomial infection in the event of a PPE shortage underscores the importance of service adaptation during an epidemic outbreak. Evidence from the 2013–2016 Ebola outbreak in West Africa suggested that the limited availability of PPE and lack of service provision adaptation made health facilities amplifiers of the spread of the disease [21]. As a result, many health facilities ended up closing due to HCW illnesses or to avoid infection at the health facility [22]. Supporting health facilities with implementation protocols to adapt services during shortages can help decrease the risk of nosocomial infection while maintaining service delivery [23].

This study has several limitations. First, we presented the general availability of PPE but did not provide insight on the adequacy of the stock level or the quality of the equipment at hand. Therefore, it cannot be assumed that facilities with supplies available would be able to ensure minimization of the risk of nosocomial infection. In addition, the timing of the phone survey may have also impacted our estimation depending on the delivery date of stocks and the prevalence of COVID-19 at the time of the survey. However, the small range of variations we observed over rounds suggests that levels of availability may have been somewhat stable over time. In addition, the country samples were not fully representative in all settings. The sampling strategy was usually to stratify by province and facility type to obtain nationally representative samples based on the master facility list provided by Ministries of Health which may not have been fully updated. These lists also compromised very few (or none) private health facilities, and we know little about what the situation in the private sector was in most countries.

## Conclusion

PPE is the last line of defense for HCWs [22, 23]. In this study, we showed that the current level of availability of the different PPE pieces was insufficient to guarantee the safety of HCWs and patients during care in many primary-level health facilities. We also identified that HCWs were not using the complete set of PPE while examining COVID-19 suspected cases, even when all the relevant pieces were available. Finally, we found that facilities were more likely to make IPC-related adaptations than changes to service delivery.

While efforts were made to accelerate and enhance the production and dissemination of PPE globally, the availability of the equipment continued to require critical attention in many LMICs. Moreover, the availability of PPE should be accompanied by communication, supportive supervision, and attention to behavior change to increase their use. Finally, we believe closer attention to the implementation of IPC measures and more assistance to help health facilities identify and interpret PPE shortages are needed to establish an actionable plan for service provision adaptation.

## Supporting information

**S1 Table. Availability of PPE at the health facilities over rounds.**
(DOCX)

**S2 Table. Sampling strategy for each country.**
(DOCX)

**S3 Table. Response rate to the health facility phone survey for the round of the study.**
(DOCX)

**S4 Table. Availability of PPE before the COVID-19 pandemic in Burkina Faso and Guinea.**
(DOCX)

## Acknowledgments

We gratefully acknowledge the contributions of Alain-Desire Karibwami, Isidore Sieleunou, Munirat Ogunlayi, Julie Ruel Bergeron and other focal points from the GFF secretariat and GFF liaison officers Jean Christian Youmba, Marie Louise Mbula, Freddy Essimbi Onana Essomba, Mardieh Dennis, Pius MasaukoNakoma, Mamadou Namory Traoré and Umma Yaraduafor for facilitating data collection.

## Author Contributions

**Conceptualization:** Salomé Henriette Paulette Drouard, Tashrik Ahmed, Gil Shapira.

**Data curation:** Salomé Henriette Paulette Drouard, Pablo Amor Fernandez.

**Formal analysis:** Salomé Henriette Paulette Drouard.

**Funding acquisition:** Tawab Hashemi.

**Investigation:** Salomé Henriette Paulette Drouard, Pablo Amor Fernandez, Prativa Baral.

**Methodology:** Salomé Henriette Paulette Drouard.

**Project administration:** Salomé Henriette Paulette Drouard, Tawab Hashemi, Isidore Sieleunou, Munirat Iyabode Ogunlayi, Alain-Desire Karibwami, Julie Ruel Bergeron.

**Software:** Salomé Henriette Paulette Drouard.

**Supervision:** Salomé Henriette Paulette Drouard.

**Validation:** Isidore Sieleunou, Munirat Iyabode Ogunlayi, Alain-Desire Karibwami, Julie Ruel Bergeron, Edwin Eduardo Montufar Velarde, Mohamed Lamine Yansane, Chea Sanford Wesseh, Charles Mwansambo, Charles Nzelu, Helal Uddin, Mahamadi Tassembedo.

**Visualization:** Salomé Henriette Paulette Drouard.

**Writing – original draft:** Salomé Henriette Paulette Drouard.

**Writing – review & editing:** Salomé Henriette Paulette Drouard, Tashrik Ahmed, Michael Peters, Peter Hansen, Gil Shapira.

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
