## [Decision Letter · Decision Letter 0]

28 Mar 2023

PONE-D-22-26371Availability and use of personal protective equipment in low- and middle-income countries during the COVID-19 pandemicPLOS ONE

Dear Dr. drouard,

Thank you for submitting your manuscript to PLOS ONE. After careful consideration, we feel that it has merit but does not fully meet PLOS ONE’s publication criteria as it currently stands. Therefore, we invite you to submit a revised version of the manuscript that addresses the points raised during the review process.

We look forward to receiving your revised manuscript.

Kind regards,

Allen Prabhaker Ugargol

Academic Editor

PLOS ONE

Journal Requirements:

4. Please include a caption for figure 1.

Additional Editor Comments (if provided):

Dear Authors,

The reviewers have provided their suggestions for improving the manuscript and the following observations are in order. We look forward to your revision as per these comments and suggestions.

1. The English used is in the present tense, while grammatically all manuscripts must be in the past tense, particuarly when it is a retrospective analysis

2. This is a survey which does not state how many times a telephone call or a contact was made and how many times was the response in the affirmative or negative

3. What were the criteria used for choosing the countries used in the survey For instance, Only Bangladesh features from SE Asia. The authors must define what were the criteria used for the same

4. The results are not adequate or complete to draw a proper conclusion from this study. The authors have stated that availability of the PPE doesnot gauarantee usage of PPE by the health care facility, a conclusion which is not adequately supported by the results of the study

Reviewers' comments:

Reviewer's Responses to Questions

**Comments to the Author**

1. Is the manuscript technically sound, and do the data support the conclusions?

Reviewer #1: Yes

Reviewer #2: Partly

2. Has the statistical analysis been performed appropriately and rigorously? 

Reviewer #1: Yes

Reviewer #2: No

3. Have the authors made all data underlying the findings in their manuscript fully available?

Reviewer #1: Yes

Reviewer #2: Yes

4. Is the manuscript presented in an intelligible fashion and written in standard English?

Reviewer #1: Yes

Reviewer #2: No

5. Review Comments to the Author

Reviewer #1: Good descriptive write up on inadequacy of PPE during pandemic. Gives readers a better understanding of situation in the low- and middle- income countries. It adds to literature database on PPE supplies.

Reviewer #2: The following observations are in order :

1. The English used is in the present tense, while grammatically all manuscripts must be in the past tense, particuarly when it is a retrospective analysis

2. This is a survey which does not state how many times a telephone call or a contact was made and how many times was the response in the affirmative or negative

3. What were the criteria used for choosing the countries used in the survey For instance, Only Bangladesh features from SE Asia. The authors must define what were the criteria used for the same

4. The results are not adequate or complete to draw a proper conclusion from this study. The authors have stated that availability of the PPE doesnot gauarantee usage of PPE by the health care facility, a conclusion which is not adequately supported by the results of the study

6. PLOS authors have the option to publish the peer review history of their article (what does this mean?). If published, this will include your full peer review and any attached files.

Reviewer #1: **Yes: **Moi Lin Ling

Reviewer #2: No

---

## [Author Response · Author response to Decision Letter 0]

24 May 2023

To the editorial board of PLOS ONE,

Thank you for the invitation to respond to your and the reviewers’ comments and revise our manuscript. The comments were very helpful in improving the writing and better presenting our methodology and communicate our findings. 

We are excited to submit a revised and improved manuscript for your review. Please find below a point-by-point response to the review comments. For easier reference, the received comments are boldfaced.

Sincerely,

Salomé Drouard

Editor’s comments

1. Please ensure that your manuscript meets PLOS ONE's style requirements, including those for file naming. The PLOS ONE style templates can be found at.

Response: Thank you. We have revised the manuscript according to the PLOS ONE’s guidelines.

Response: Thank you for the clarification, we added the following paragraph in the method section.

In the data collection paragraph: “Survey respondents generally included facility officer in-charges, but in some cases other respondents, like facility administrators, were better suited to answer modules within the survey. 

In the ethical approval paragraph: “Survey participation was voluntary and verbal consent was received from all respondents.”

Response: We added the following paragraph in the ethical approval paragraph of the method section.

“The study was requested, reviewed, and approved by a director-level official in each Ministry of Health and was exempted from human subjects research as public health practice in every country except Burkina Faso. In Burkina Faso, ethical approval was received from the ethics committee of the local author’s institute.” 

4. Please include a caption for figure 1.

Response: Thank you. The caption was added to the main text with a legend. 

Response: The captions were added, and the supporting information file updated according to the guidelines. 

Response: Thank you for your careful review and we apologize for this discrepancy due to an error in importing the bibliography from Mendeley. The following references were added in this order. 

1. Cancedda C, Davis SM, DIerberg KL, Lascher J, Kelly JD, Barrie MB, et al. Strengthening Health Systems While Responding to a Health Crisis: Lessons Learned by a Nongovernmental Organization During the Ebola Virus Disease Epidemic in Sierra Leone. J Infect Dis. 2016;214: S153–S163. doi:10.1093/INFDIS/JIW345

2. Reddy SC, Valderrama AL, Kuhar DT. Improving the Use of Personal Protective Equipment: Applying Lessons Learned. Clinical Infectious Diseases. 2019;69: S165–S170. doi:10.1093/CID/CIZ619

3. Keep health workers safe to keep patients safe: WHO. [cited 28 Nov 2021]. Available: https://www.who.int/news/item/17-09-2020-keep-health-workers-safe-to-keep-patients-safe-who

4. Second round of the national pulse survey on continuity of essential health services during the COVID-19 pandemic. [cited 28 Nov 2021]. Available: https://www.who.int/publications/i/item/WHO-2019-nCoV-EHS-continuity-survey-2021.1

5. Kazungu J, Munge K, Werner K, Risko N, Vecino-Ortiz AI, Were V. Examining the cost-effectiveness of personal protective equipment for formal healthcare workers in Kenya during the COVID-19 pandemic. BMC Health Serv Res. 2021;21: 1–7. doi:10.1186/S12913-021-07015-W/FIGURES/5

Reviewer 1’s comments 

1. The English used is in the present tense, while grammatically all manuscripts must be in the past tense, particuarly when it is a retrospective analysis.

Response: Thank you for pointing this out. The manuscript was re-written in past tense.

2. This is a survey which does not state how many times a telephone call or a contact was made and how many times was the response in the affirmative or negative.

Response: Thank you for this comment. We added the following extract to the data collection paragraph in the method section and the following table to the supporting information:

Three attempts were made to reach each facility, and interview times were scheduled in advance to minimize burden on the respondents. In case of non-response, a replacement facility of the same facility level in the same province was randomly selected from the list of eligible health facilities when possible. More details on the response rate are available in S3 Table in the appendix. All the health facility representatives we managed to reach accepted to take part in the survey. 

S3 Table. Response rate to the health facility phone survey for the round of the study

Country Round # of health facilities selected # of health facilities interviewed # of health facilities who cannot be reached or replaced Response rate

Bangladesh Jul-21 300 291 9 98%

Burkina Faso Aug-21 159 159 0 100%

Guatemala Jun-21 255 239 16 94%

Guinea Jul-21 160 156 4 98%

Liberia Jul-21 122 116 6 97%

Malawi Jun-21 204 192 12 94%

Nigeria May-21 421 401 20 95%

3. What were the criteria used for choosing the countries used in the survey For instance, Only Bangladesh features from SE Asia. The authors must define what were the criteria used for the same

Response: This study was part of a broader initiative supported by the Global Financing Facility for Women, Children, and Adolescents (GFF) to help member countries to monitor the effect of the COVID-19 pandemic on essential health services. All GFF member countries were offered to participate in the survey. The countries covered in this article were the countries for which at least one round of data was collected by August 2021.

We added the following extract to the overview and sample selection paragraph in the method section to document that point:

“In this context, implementation of facility phone surveys was offered to all partner countries. The seven countries covered by this study are the ones that opted to implement the phone survey and for which at least one round of data was completed by August 2021.” 

4. The results are not adequate or complete to draw a proper conclusion from this study. The authors have stated that availability of the PPE does not guarantee usage of PPE by the health care facility, a conclusion which is not adequately supported by the results of the study. 

Response: As we highlight in our paper, the availability of all the PPE barriers at the health facility does not guarantee that HCWs will wear all the recommended items when examining patients suspected to be infected with COVID-19. However, while not sufficient, the availability of all barrier PPE is necessary to comply with infection prevention control protocols. 

We believe that our paper makes that point by stressing that, when restricting the sample to health facilities with all PPE barriers available, only 61% of the respondents declared their HCWs were wearing the complete PPE set when examining potential COVID-19 cases. 

The results of our study imply that there is not only an urgent need to improve the availability of PPE barriers, as emphasized in the first part the article, but there is also a need for additional interventions to ensure HCWs appropriately use the barriers when they are available.

To make this point clearer we rephrase the following sentences in the result and discussion sections.

Rephrased sentences the result section:

“We also found that HCWs did not use all the recommended PPE barriers when examining COVID-19 suspected or confirmed cases even when all items are available at the health facility. Restricting the sample to only health facilities with the complete PPE set available, we found that HCWs were wearing all the recommended barriers in only 61% of health facilities.” 

Rephrased sentence the discussion section:

“We also found that HCWs examining COVID-19 suspected cases were not systematically using the complete PPE set even when all barriers were available in their facilities.”

---

## [Editor Report · Decision Letter 1]

28 Jun 2023

Availability and use of personal protective equipment in low- and middle-income countries during the COVID-19 pandemic

PONE-D-22-26371R1

Dear Dr. Drouard,

We’re pleased to inform you that your manuscript has been judged scientifically suitable for publication and will be formally accepted for publication once it meets all outstanding technical requirements.

Kind regards,

Allen Prabhaker Ugargol

Academic Editor

PLOS ONE
---

## [Editor Report · Acceptance letter]

7 Jul 2023

PONE-D-22-26371R1 

Availability and use of personal protective equipment in low- and middle-income countries during the COVID-19 pandemic 

Dear Dr. Drouard:

I'm pleased to inform you that your manuscript has been deemed suitable for publication in PLOS ONE. Congratulations! Your manuscript is now with our production department. 

Kind regards, 

on behalf of

Dr. Allen Prabhaker Ugargol 

Academic Editor

PLOS ONE